Citation: *Molecular Systems Biology* 9:709
www.molecularsystemsbiology.com

# Transcriptional regulation is insufficient to explain substrate-induced flux changes in *Bacillus subtilis*

Victor Chubukov[1,4], Markus Uhr[2,4], Ludovic Le Chat[3,4], Roelco J Kleijn[1,4], Matthieu Jules[3], Hannes Link[1], Stephane Aymerich[3], Jörg Stelling[2] and Uwe Sauer[1,*]

[1] Institute of Molecular System Biology, ETH Zurich, Zurich, Switzerland, [2] Department of Biosystems Science and Engineering, SIB Swiss Institute of Bioinformatics, ETH Zurich, Zurich, Switzerland and [3] Micalis Institute, INRA, AgroParisTech, Thiverval-Grignon, France
[4]These authors contributed equally to this work.
* Corresponding author. Institute of Molecular Systems Biology, ETH Zurich, Wolfgang Pauli Strasse 16, Zurich CH-8093, Switzerland. Tel.: + 41 44 633 3672; Fax: + 41 44 633 1051; E-mail: sauer@ethz.ch

One of the key ways in which microbes are thought to regulate their metabolism is by modulating the availability of enzymes through transcriptional regulation. However, the limited success of efforts to manipulate metabolic fluxes by rewiring the transcriptional network has cast doubt on the idea that transcript abundance controls metabolic fluxes. In this study, we investigate control of metabolic flux in the model bacterium *Bacillus subtilis* by quantifying fluxes, transcripts, and metabolites in eight metabolic states enforced by different environmental conditions. We find that most enzymes whose flux switches between on and off states, such as those involved in substrate uptake, exhibit large corresponding transcriptional changes. However, for the majority of enzymes in central metabolism, enzyme concentrations were insufficient to explain the observed fluxes— only for a number of reactions in the tricarboxylic acid cycle were enzyme changes approximately proportional to flux changes. Surprisingly, substrate changes revealed by metabolomics were also insufficient to explain observed fluxes, leaving a large role for allosteric regulation and enzyme modification in the control of metabolic fluxes.

*Molecular Systems Biology* **9**: 709; published online 26 November 2013; doi:10.1038/msb.2013.66
*Subject Categories:* metabolic and regulatory networks; cellular metabolism
*Keywords:* central carbon metabolism; metabolic flux; transcriptional regulation

## Introduction

A key feature of a microorganism's environment is the presence or absence of metabolizable substrates. Heterotrophic bacteria are able to consume a variety of carbon sources for growth, but to do this, they must rearrange their metabolic programs to allow for the variety of metabolic flux patterns (Kleijn *et al*, 2010; Beste *et al*, 2011). This can be accomplished by various means, including thermodynamic effects through changes in metabolite concentrations or modulation of enzyme activity through protein modifications or allosteric regulation by small molecules (Fonseca *et al*, 2011; Gerosa and Sauer, 2011). However, the most well-studied case is the alteration of enzyme concentration through transcriptional regulation. This attention is driven by many canonical examples of enzyme induction in response to a rising demand for flux, such as the induction of the lac operon in response to lactose availability (Jacob and Monod, 1961), or the induction of an amino-acid biosynthesis pathway in response to depletion of the amino acid in the medium (Zaslaver *et al*, 2004; Chubukov *et al*, 2012). Such examples have created an appealing intuitive picture: flux is primarily controlled by the availability of enzymes. However, when the changes in flux are small compared with the hundred-fold or thousand-fold

changes in the examples above, this intuitive picture breaks down. This is seen in a number of findings, for instance, the lack of glycolytic flux changes upon overexpression of many yeast glycolysis enzymes (Hauf *et al*, 2000), or the lack of changes in the flux distribution upon deletion of any of a number of seemingly important transcription factors in *Saccharomyces cerevisiae* (Fendt *et al*, 2010) or *E. coli* (Haverkorn van Rijsewijk *et al*, 2011). Those results suggest a contrary picture, where enzyme expression through transcriptional regulation is not crucial for control of flux.

A further motivation for understanding the relationship between transcriptional regulation and metabolic phenotype is the interpretation of gene expression data. With the increased standardization of high-throughput transcriptomics methods (Slonim and Yanai, 2009; Wang *et al*, 2009), quantifying gene expression changes in response to environmental changes has become commonplace. Because expression of many metabolic enzymes (e.g., the aforementioned sugar utilization and amino acid biosynthesis enzymes) is under the control of transcriptional regulators that can sense relevant environmental signals (Wall *et al*, 2004; Seshasayee *et al*, 2009), it is tempting to interpret most enzyme expression changes as changes in the metabolic phenotype, that is, changes in flux.

However, if transcript levels are not in fact controlling fluxes, such an interpretation will be misleading. In that case, transcript changes may simply be due to crosstalk or suboptimal gene regulation (Price *et al*, 2013), while flux would be controlled at other levels, such as substrate availability, allosteric regulation, enzyme modifications, or translational control of enzyme expression.

In this study, we take a systems-level view of the mechanisms behind changes in metabolism in the model Gram-positive bacterium *Bacillus subtilis* by quantifying fluxes, transcripts, and metabolites in eight metabolic states enforced by different environmental conditions. While previous studies have attempted to quantify the contribution of transcriptional regulation to flux changes in many model organisms (Ter Kuile and Westerhoff, 2001; Even *et al*, 2003; Rossell *et al*, 2005, 2006, 2008; Brink *et al*, 2008; Postmus *et al*, 2008), they have typically relied on flux values derived solely from uptake and secretion rates, and have often considered only pairwise comparisons among conditions. Here, we base our analysis on higher confidence flux measurements from isotopic labeling experiments and we extend the computational framework to consider a large range of environmental conditions concurrently. Finally, using quantitative data on metabolite concentrations, we are able to assess the contribution of substrate changes to metabolic flux and thus form more detailed hypotheses regarding the control of flux at each reaction.

## Results

### Inference of metabolic fluxes from isotopic labeling and enzyme concentration from transcriptomics

To analyze the contribution of transcriptional regulation to metabolic flux adjustments, we quantified both fluxes and transcripts under conditions that led to differences in metabolic fluxes. We chose environments composed of eight different combinations of carbon sources that enter metabolism at different points and allow for a range of growth rates between 0.22 and 0.75 h$^{-1}$ (Figure 1). We inferred metabolic

fluxes from $^{13}$C-labeling experiments using a comprehensive isotopomer balancing model (Van Winden *et al*, 2005; Zamboni *et al*, 2009) (Supplementary Table S4). The fluxes were indeed highly variable: of the 28 non-collinear fluxes that were non-zero in more than one condition, 25 showed at least a two-fold change between the minimum and maximum values, and 17 showed at least a five-fold change. For the three conditions where fluxes had been previously analyzed (Kleijn *et al*, 2010), we found excellent quantitative agreement between our data and previously published results. An unexpected finding was that with the exception of the gluconate condition, we consistently observed back fluxes from the tricarboxylic acid (TCA) cycle into lower glycolysis via PEP carboxykinase and/or malic enzyme. A large portion of these backfluxes were channeled back into the TCA cycle, an effect most pronounced on substrates that feed directly into the TCA cycle, such as malate and succinate + glutamate (Figure 1).

Transcript abundances in the same eight conditions were quantified using whole genome tiling arrays. While the major differences between conditions correlated with growth rate changes (Supplementary Figure S1), we also observed several surprising gene expression patterns, such as the upregulation of a number of stress response genes and a large but incomplete set of sporulation genes during growth on pyruvate or succinate + glutamate, the two slowest growth conditions in our study. This suggests that slow growth may induce some responses that mimic starvation, perhaps through crosstalk of the corresponding regulatory programs (Supplementary Figure S2; Supplementary Tables S1 and S2).

### Extension of regulatory analysis to quantify contribution of transcriptional changes to flux

To determine the contribution of the observed transcript changes to the observed flux changes, we develop a mathematical framework based on regulatory analysis (Van Eunen *et al*, 2011). The first step is to estimate enzyme concentrations based on the transcript abundance. While absolute quantification requires knowledge of individual

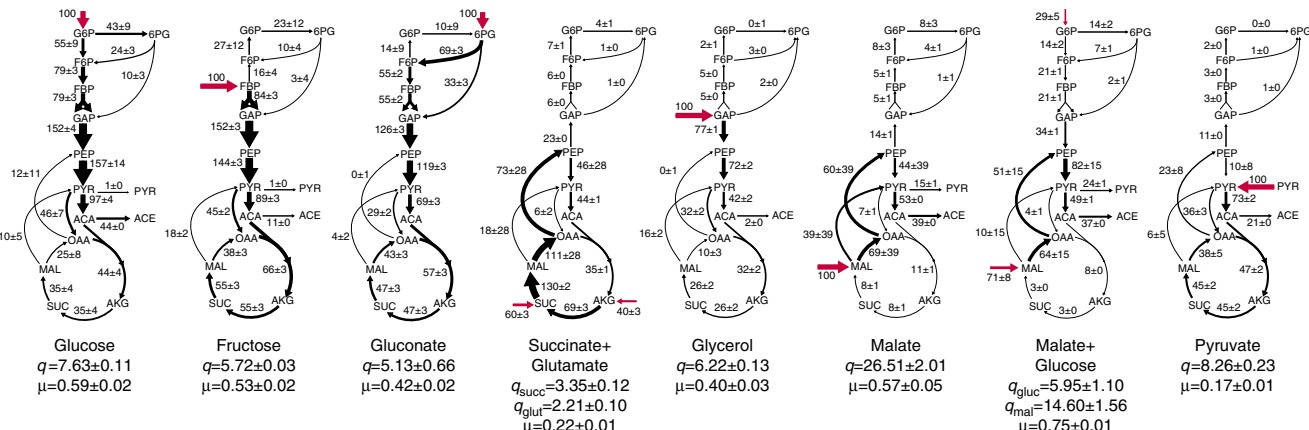

**Figure 1** Fluxes through *B. subtilis* central metabolism under eight conditions defined by different carbon sources. Numbers and sizes of arrows are normalized to the substrate uptake rate in each condition. For further analysis, absolute fluxes (mmol h$^{-1}$ gcdw$^{-1}$) were used. Substrate uptake rates (*q*) are given in mmol h$^{-1}$ gcdw$^{-1}$ and growth rates (μ) are in h$^{-1}$.

mRNA translation and decay rates, calculation of relative enzyme concentrations between conditions can be done simply based on the assumption that translation rates are not affected by the environmental perturbations. The enzyme concentrations can be written as a simple function of the measured transcript levels and growth rates:

$$E_{ij} = \alpha m_{ij} \frac{[\text{total mRNA}]_j}{\mu_j}$$

where the index $i$ refers to one of the enzymes in the metabolic network and the index $j$ to one of the eight conditions. $m_{ij}$ is the measured tiling array signal for the corresponding mRNA, [total mRNA] is the amount of mRNA per unit of biomass, $\mu$ is the growth rate, and $\alpha$ is an arbitrary constant. The amount of mRNA per biomass was assumed to be a constant fraction of total RNA, while total RNA was assumed to be a function of the growth rate; an affine fit was calculated based on the previous data (Dauner *et al*, 2001).

Relating enzyme concentration to flux requires some assumptions about enzyme kinetics. However, virtually all models of enzyme kinetics allow the decomposition of flux into the contribution of enzyme concentration, which is linear, and the possibly non-linear contribution of metabolite (substrate, product, and effector) concentrations. For instance in the case of irreversible Michaelis–Menten kinetics

$$J_{ij} = E_{ij} v_i \left( \frac{S_{ij}}{S_{ij} + k_i} \right)$$

where $J$ represents the flux, $E$ and $S$ the enzyme and substrate concentrations, and $k$ and $v$ the binding constant and turnover rate, respectively. More generally, for almost any model of enzyme kinetics, we can write

$$J_{ij} = E_{ij} v_i f_i(M_j)$$

where $M$ represents the metabolic state of the cell (i.e., the concentrations of all metabolites including substrates, cofactors, activators, and inhibitors). To analyze the relative flux and enzyme levels between two conditions as a linear problem, we move to log space

$$\Delta \log(J_i) = \Delta \log(E_i) + \Delta \log(f_i(M))$$

and define the relative contributions of enzyme concentration, $\rho_h$, and that of the metabolic state $\rho_m$ following the notation of Ter Kuile and Westerhoff (2001).

$$\rho_{h_i} = \frac{\Delta \log(E_i)}{\Delta \log(J_i)}$$

$$\rho_{m_i} = 1 - \rho_{h_i} = \frac{\Delta \log(f_i(M))}{\Delta \log(J_i)}$$

While estimating $\rho_m$ directly is impossible without knowing the function $f(M)$, we can directly estimate $\rho_h$ by quantifying flux and enzyme levels. Such analysis has been performed previously using flux and enzyme measurements in two different conditions (Rossell *et al*, 2006; Daran-Lapujade *et al*, 2007; Postmus *et al*, 2008; Van Eunen *et al*, 2009). However, pairwise comparisons lead to a number of issues, such as the direct propagation of measurement errors into the estimate of the regulatory coefficient. Here, we develop a more robust approach, which considers the flux and enzyme levels across a

spectrum of conditions at once, thus leveraging more data to estimate the regulatory coefficient. If across all the condition changes, the contribution of enzyme changes to flux is the same, then one can estimate $\rho_h$ by simply fitting a linear function

$$\log(E_i) = \rho_{h_i} \log(J_i) + \beta$$

where the constant $\beta$ depends on the units of $E$ and $J$ and can be eliminated by normalization to the mean. While such linear fitting has been used to estimate $\rho_h$ previously from slight environmental perturbations (Ter Kuile and Westerhoff, 2001; Even *et al*, 2003), here we apply it to the large flux changes across our eight conditions to ask if enzyme and flux changes consistently correlated across the spectrum of growth environments. For this analysis, we consider the basic hypotheses summarized in Figure 2.

When $\rho_h = 1$ is a good fit for a particular reaction (Figure 2B), enzyme concentration changes are faithfully reflected in flux changes. Because large deviations in central metabolic fluxes are likely to lead to fitness defects (Fischer and Sauer, 2003; Haverkorn van Rijsewijk *et al*, 2011), one parsimonious inference is that such enzymes are likely subject to particularly precise control at the gene expression level. However, $\rho_h = 1$ does not guarantee that perturbations in enzyme level will lead to changes in flux nor does it guarantee that these enzymes are not present in excess. Flux could still be controlled by unmeasured metabolic changes, while transcriptional changes match flux changes either purely by chance or as a means of keeping metabolite concentrations constant

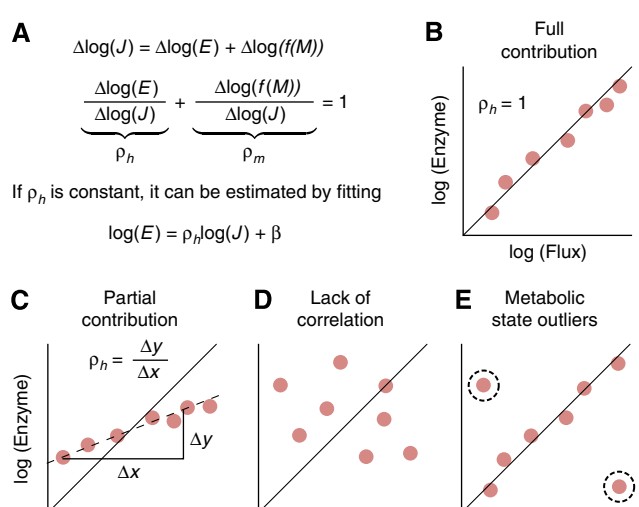

**Figure 2** Basic hypotheses of regulatory analysis. (**A**) Flux ($J$) changes can be decomposed into the contribution of enzyme concentration ($E$) changes and changes in the metabolic state ($M$). If the relative contributions of the two are constant, then the contribution of enzyme levels, $\rho_h$, can be estimated by the linear fit of a log-log plot. If $\rho_h$ is near 1 (**B**), then it is consistent with changes in enzyme levels being entirely responsible for observed changes in flux. Otherwise, despite good correlation between flux and enzyme (**C**), other mechanisms such as substrate concentration changes, allosteric regulation, or enzyme modification are necessary to explain flux changes. Other possible outcomes could be incoherent flux and enzyme changes (**D**) or generally good agreement between enzyme concentration and flux but with several conditions where a distinct metabolic state is reached, which would show up as outliers from an otherwise good fit (**E**).

across conditions (Kacser and Acerenza, 1993; Fell and Thomas, 1995).

Nevertheless, when changes in metabolite abundances are responsible for the majority of changes in flux, enzyme concentration will generally be a poor predictor of flux, showing either a poor fit or a $\rho_h$ close to zero (Figure 2C and D). In such cases, transcriptional control of flux across the spectrum of conditions can be excluded, with the data instead pointing to consistent excess of enzyme, and flux regulation through metabolite concentrations or enzyme modification. It also need not be the case that $\rho_h$ is constant across all conditions. If enzyme concentration is generally the driver of flux through a reaction, but a particularly different metabolic state that strongly affects the flux is reached under one growth condition, then that point will be an outlier from the linear fit (Figure 2E).

## Regulatory analysis reveals a significant contribution of transcriptional regulation only for a small set of reactions

To determine the extent of transcriptional regulation in *B. subtilis* central metabolism according to the analysis outlined above, we paired each of the inferred central metabolic fluxes with every enzyme that can catalyze the corresponding reaction. Limiting the analysis to fluxes that were non-zero (greater than an absolute cutoff of 0.1 mmol gcdw$^{-1}$ h$^{-1}$) in at least three conditions led to 46 pairwise comparisons where we could examine the relationship between flux changes and enzyme concentration changes. In order to restrict the influence of outliers on the linear fitting, we use the weighted Theil-Sen estimator (Jaeckel, 1972; Birkes and Dodge, 1993), which considers the median of the slopes through all pairs of points, instead of the traditional least-squares method. A confidence interval for $\rho_h$ was obtained by resampling random subsets of the data and concurrently perturbing both the flux and the enzyme data according to their measurement error.

We obtained a wide range of patterns for the various enzymes in central carbon metabolism. Illustrations of four key patterns are depicted in Figure 3A–D. For instance, succinyl-CoA synthetase (SucC) showed a clear correlation between enzyme and flux as well as a $\rho_h$ near 1 (Figure 3A). This is consistent with the hypothesis that the cell adjusts this flux in response to different environments solely by manipulating the enzyme level. Other enzymes, such as phosphoglycerate mutase (Pgm), despite a high correlation between enzyme and flux, showed $\rho_h$ far below 1 (Figure 3B), meaning that while enzyme changes likely contribute to control of flux, they alone cannot explain the changes in flux across conditions. In other cases, there was no correlation between enzyme and flux levels, either when enzyme concentrations were essentially constant despite large changes in flux, as was the case for glucose-6-phosphate dehydrogenase (Zwf) (Figure 3C), or when flux and enzyme changes were incoherent across conditions, such as for the secondary isoform of citrate synthase (CitA) (Figure 3D). In those cases, unless other enzymes can catalyze the same reaction, metabolic effects such as substrate availability, allosteric regulation, or enzyme modification must explain the vast majority of flux changes.

Neighboring metabolic reactions showed similar $\rho_h$ estimates (Figure 3F). Most instances of reactions consistent with strong control of flux by enzyme concentration were in the TCA cycle—in fact all four enzymes for which the data showed a good linear fit and the estimate of $\rho_h$ was not statistically distinguishable from 1 (at $\alpha = 0.05$) were in this pathway: SucC, SucD, Icd, and CitB (Figure 3E). Since one of the key features of transcriptionally controlled reactions is that enzymes are not expressed at higher than necessary levels, this could potentially indicate that these enzymes have a high cost of expression that the cell tries to minimize. Meanwhile, some of the weakest correlations between enzyme and flux were found among pentose phosphate pathway enzymes. This indication of significant metabolic regulation could be explained by the necessity to quickly change flux through this pathway in response to conditions such as oxidative stress, as has already been demonstrated in yeast (Ralser *et al*, 2009).

The most striking result is that very few reactions are consistent with full control of flux by enzyme levels, that is, that in general, enzymes are available in excess and other mechanisms are responsible for controlling exact flux magnitudes *in vivo*. This opens the question of which alternative mechanisms are responsible for modifying flux in response to environmental changes. To further quantify the other possible contributions to flux control, we examined indirect control by other enzymes, and metabolite-level control by changing substrate concentrations.

## Direct analysis of metabolic split ratios

One possible explanation of how transcriptional regulation could still have a major influence on fluxes despite the apparent lack of proportional changes between fluxes and enzyme concentrations is through the control of upstream steps in a linear pathway. Flux changes through one or more enzymes might then propagate down the pathway by intermediate substrate accumulation without quantitative agreement between fluxes and enzyme concentrations in subsequent steps of the pathway, as long there was sufficient enzyme expression. However, whenever a branch point in metabolism is reached, there is no such indirect control by other enzymes, and the fraction of flux diverted to each branch depends only on the local enzyme kinetics (assuming irreversible reactions and no product inhibition). As such, a more direct test of control of flux by transcriptional regulation comes from analyzing the split ratio at a particular branch point and relating it to the concentrations of the enzymes catalyzing the reaction on each branch. To consider this quantitatively, we constructed a simple mathematical model to calculate this split ratio as a function of enzyme concentrations for the case when one substrate can be converted by one of two irreversible enzymes, and scanned a large range of values for the enzyme kinetic parameters to determine the relationship between the enzyme concentration ratio and the flux split ratio, again assuming Michaelis–Menten kinetics and ignoring product inhibition. We find that for a wide range of parameters, the change in flux ratio between two conditions will be exactly equal to the change in the enzyme concentration ratio. This relationship only

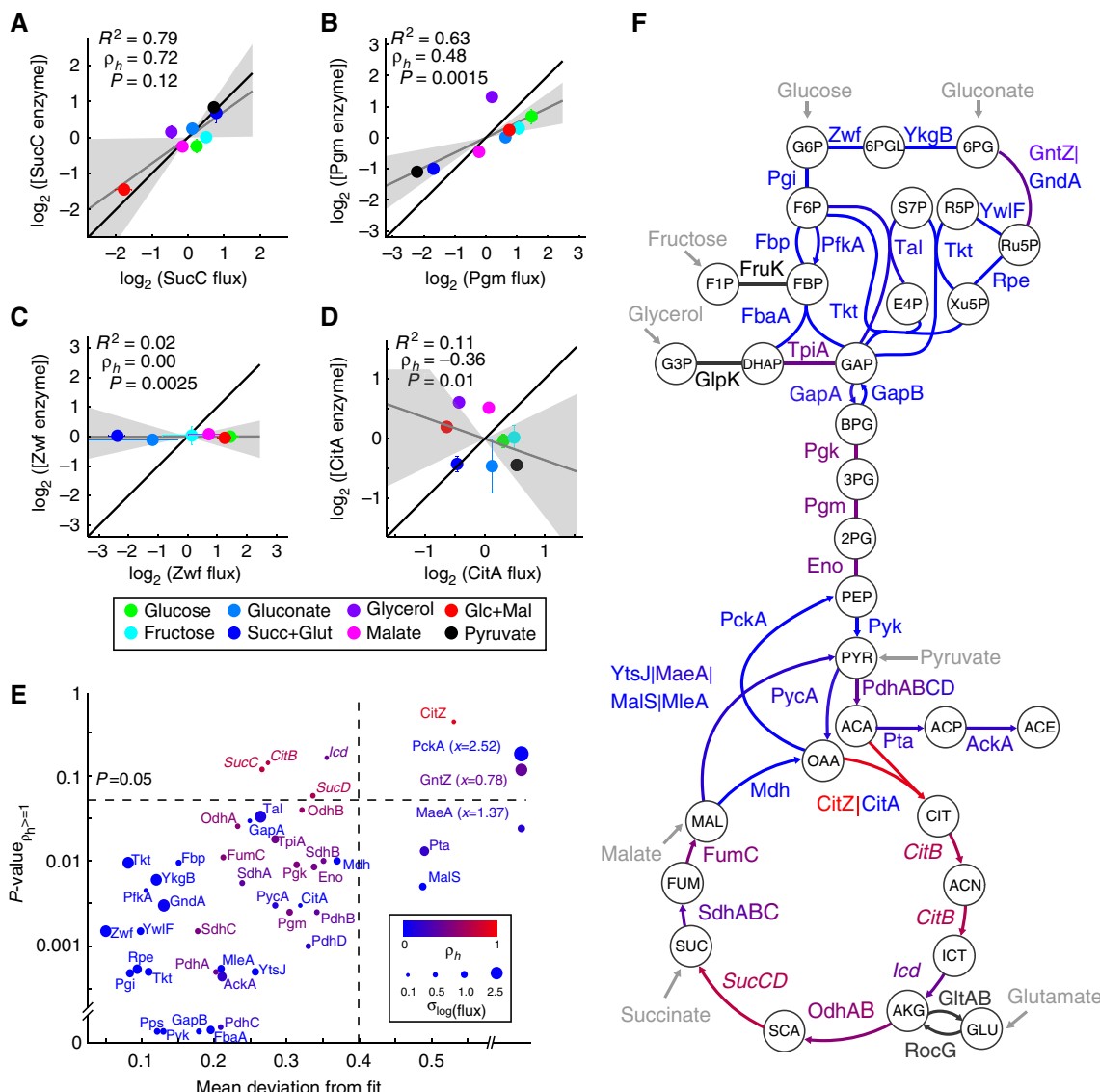

**Figure 3** Correlation of flux and enzyme concentration (inferred from transcripts) changes for reactions in central metabolism. Four typical cases are shown in (**A–D**). Both X and Y data are $\log_2$-transformed and normalized by subtracting the mean, forcing the line of best fit to pass through the origin. For each plot, the coefficient of determination ($R^2$) and the slope ($\rho_h$) of the line of best fit (gray) are given, along with $P$, the $P$-value associated with the hypothesis $\rho_h > 1$ (see Materials and methods). The shaded area corresponds to a 95% confidence interval of the fitted slope and the black line corresponds to $\rho_h = 1$. (**E**) Scatter plot of $P$-values and goodness of linear fit for each reaction. To distinguish cases like (**C**) from (**D**), both of which give $R^2 \approx 0$, we take the mean value of the residuals as the goodness of fit for the purpose of this plot. Symbol size corresponds to the deviation in flux values (considering only fluxes greater than the cutoff of 0.1 mmol gcdw$^{-1}$ h$^{-1}$). Color corresponds to $\rho_h$ values (the slope of the fitted line). Enzymes where $P < 0.05$ (i.e., $\rho_h = 1$ cannot be statistically excluded) and the mean residual value is $< 0.4$ (i.e., good fit) are shown in italic. Some points that overlap perfectly have been offset slightly for visibility. (**F**) Graphical representation of $\rho_h$ values for each enzyme in central carbon metabolism. Color legend and fonts are identical to (**E**).

breaks down for extreme cases, such as when one enzyme has orders of magnitude lower affinity (high $k_m$) and compensates with higher $\nu_{max}$ (concentration times catalytic rate) (Supplementary Figure S3).

Because of the mostly linear relationship between flux ratio and enzyme ratio, we can apply the analysis developed in the previous section to determine whether flux splits in *B. subtilis* central carbon metabolism are likely to be transcriptionally controlled. We considered branch points where we could correlate at least five data points and we eliminated the split involving malic enzyme because of the high uncertainty in

the estimation of this flux (Figure 4). We find a clearly non-monotonic relationship for the split at oxaloacetate between citrate synthase leading to the TCA cycle and gluconeogenic PEP carboxykinase, meaning that this split ratio cannot be controlled at the transcriptional level between our conditions. For the two other branch points, at pyruvate between pyruvate decarboxylase and pyruvate dehydrogenase, and at acetyl-CoA between the TCA cycle and acetate production, the relationship between the flux and enzyme ratio was generally monotonic (one significant outlier in each case), and we found a reasonable fit to a linear relationship.

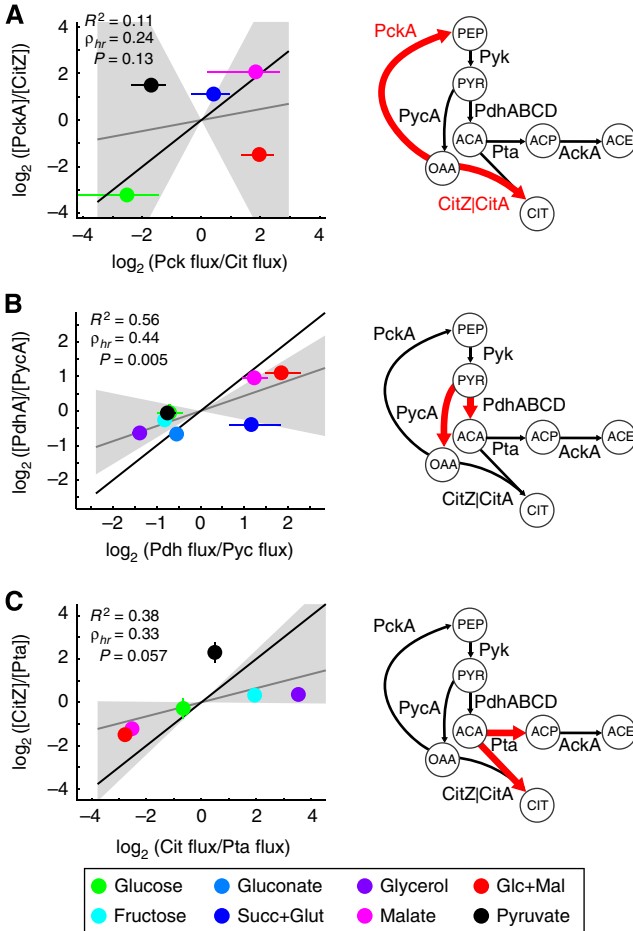

**Figure 4** Ratio between flux split ratios and enzyme concentration ratios at three branch points in central metabolism (**A–C**). As in Figure 3, the coefficient of determination ($R^2$) and slope of the line of best fit ($\rho_{hr}$) are given for each plot, along with a P-value for the hypothesis $\rho_{hr} > 1$. The shaded region indicates a 95% confidence interval of the fitting and the black line corresponds to slope 1.

However, the slope of the inferred fit, which we call $\rho_{hr}$, was significantly below 1, meaning that changes in the flux ratio between conditions were larger than changes in the enzyme ratio. Under the assumptions above, this implies that mechanisms other than the control of enzyme concentration through transcriptional regulation are responsible for control of flux, even in cases where indirect control by other enzymes (e.g., relief of a rate-limiting step) is ruled out.

We note that our simple model assumes only one enzyme for each branch of the split, whereas, for instance, citrate synthase occurs as two isozymes. While CitZ is thought to be responsible for the majority of the activity (Jin and Sonenshein, 1994), we nevertheless sought to account for the possibility that the combination of the two isozymes CitA and CitZ drives flux changes. We performed the same analysis using various linear combinations of CitA and CitZ to calculate the enzyme ratio, corresponding to their unknown relative contributions to the flux. None of the linear combinations led to a better fit between enzyme ratios and flux ratios, confirming that

other mechanisms must regulate the flux ratio at the acetyl-CoA node.

## Changes in substrate concentration are responsible for a minority of metabolic regulation

The central result from the regulatory analysis above is that most reactions in central carbon metabolism are consistent with a minority contribution of transcriptional regulation to control of flux. To further elucidate the remaining contribution, which may come from allosteric regulation, post-transcriptional modifications, or substrate concentration changes, we quantified metabolite concentrations in all eight conditions using mass spectrometry. If changes in flux are driven by increased substrate concentration, then flux changes should correlate with changes in substrate levels. Furthermore, metabolites may regulate flux in distant reactions through allosteric interactions, and we can also investigate these distant correlations.

We quantified 35 metabolites largely consisting of intermediates of central carbon metabolism, 19 of which were substrates of the previously analyzed reactions (Supplementary Table S3). For most metabolites and conditions, accurate quantification with <10% standard deviation was possible. Some of the largest differences between conditions were seen in the TCA cycle, with for instance, fumarate and succinate exhibiting over 20-fold higher concentrations during growth on malate compared with growth on glycolytic carbon sources. In contrast, concentrations of cofactors such as ATP and intermediates of glycolysis and the pentose phosphate pathway showed lower variation. One striking phenomenon we observed was the high correlation between fructose-bis-phosphate (FBP) concentration and the magnitude of the glycolytic flux. This correlation, which was stronger for FBP than any other metabolite, suggests that FBP may have a role in sensing glycolytic flux magnitude in *B. subtilis*, similarly to its recently proposed role in *E. coli* (Kochanowski *et al*, 2013b). The fact that in *B. subtilis*, FBP negatively modulates the activity of CggR, the transcriptional repressor of lower glycolytic genes (Doan and Aymerich, 2003; Zorrilla *et al*, 2007), parallels the FBP-Cra interaction in *E. coli* and further points to conservation of this regulatory motif.

To determine whether the observed metabolite changes could explain flux changes through modulation of substrate concentrations, we derived a simplified model for the quantitative relationship between the two. In contrast to the relationship between flux and enzyme concentration, which is linear under virtually any model of enzyme kinetics, the relationship between substrate concentration and flux is generally non-linear. As such, we allowed for a free parameter to represent the contribution of substrate concentration to flux, hypothesizing a relationship of the form $\Delta J = \Delta E \Delta S^\lambda$ where the parameter $\lambda$ would represent, for example, saturation ($\lambda < 1$) or cooperativity ($\lambda > 1$). This formulation is identical to the commonly used S-system approach for modeling metabolic networks (Savageau and Voit, 1987). The optimal value for $\lambda$ was found by least squares fitting of the line $\log(J) - \log(E) = \lambda \log(S)$. The remaining analysis paralleled the earlier analysis of the flux–enzyme relationship and we introduce the

corresponding coefficient $\rho_{es}$ to represent the combined contribution of enzyme and substrate concentrations to flux.

A few reactions showed improved fit when changes in substrate concentration were taken into account. Two such cases are shown in Figure 5A and B. While enolase (Eno) was expressed at almost constant levels between the three gluconeogenic conditions, the concentration of its substrate PEP changed ~4-fold and was likely the driver of flux changes

through this reaction (Figure 5A). In another example, aconitase (CitB) enzyme levels were somewhat consistent with fluxes ($\rho_h = 0.70$, $R^2 = 0.41$), but with mild outliers corresponding to the glucose and fructose conditions (Figure 5B). The inclusion of the substrate (citrate) concentration was able to reduce the error in the fitting ($\rho_{es} = 0.67$, $R^2 = 0.55$) as growth on glucose or fructose led to relatively high citrate concentrations. However, very surprisingly, for

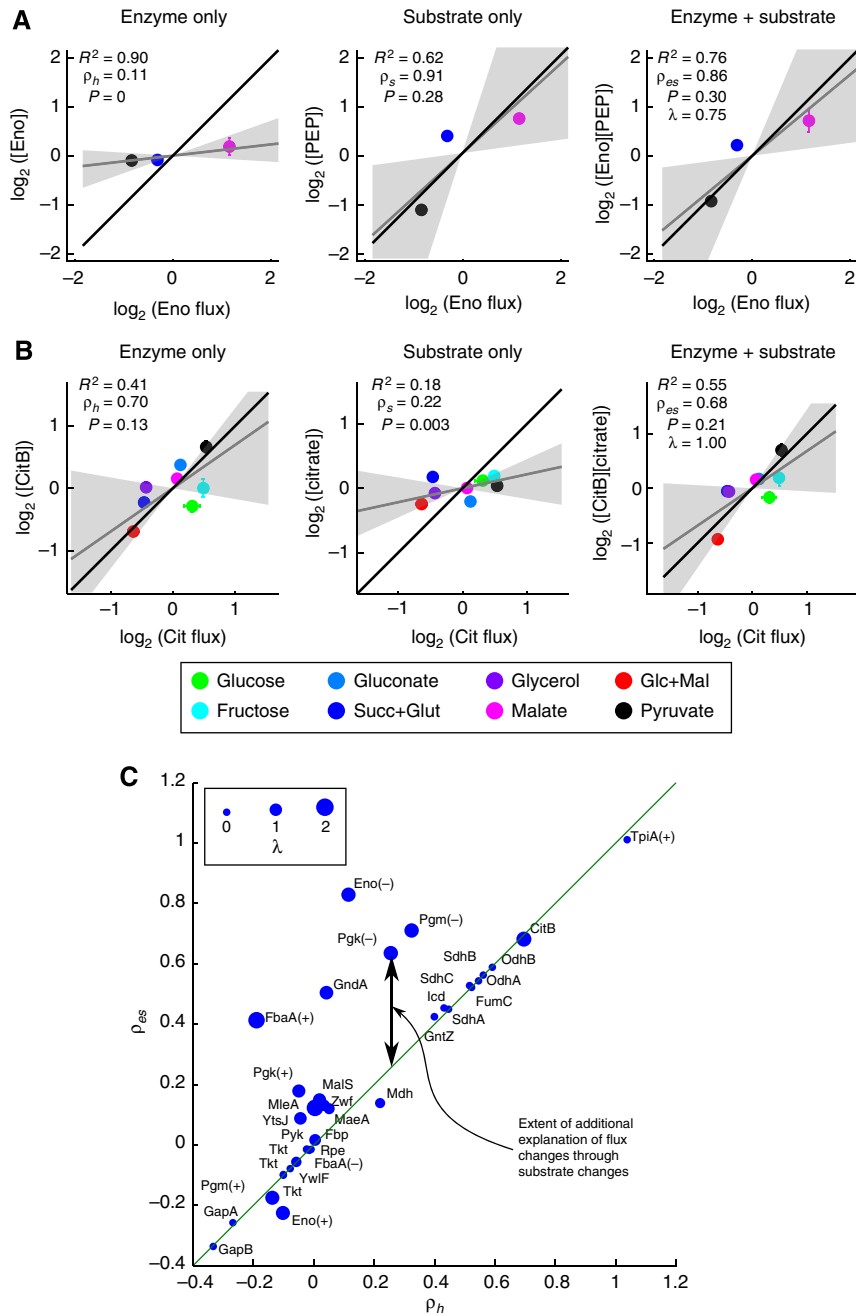

**Figure 5** Combined effects of substrate and enzyme changes on flux predictions. Substrate changes may be the dominant drivers of flux (**A**) or partially explain outliers (**B**). See Figures 3 and 4 for an explanation of the plots. $\lambda$ is the non-linearity parameter for the effect of substrate on flux ($\lambda = 1$ corresponds to an inferred linear relationship between substrate and flux). (**C**) Effect of including substrate information for all reactions. For each enzyme where substrate concentration information was available, the $\rho_h$ and $\rho_{es}$ values are shown. The difference $\rho_{es} - \rho_h$ represents the additional contribution of substrate information, which was significant only for a few reactions. Symbol size represents $\lambda$ (see above). Reversible reactions in lower glycolysis were analyzed separately for each direction and are marked with ($+$) for glycolytic and ($-$) for gluconeogenic directions. Transketolase (Tkt) appears multiple times corresponding to its multiple reactions involving different substrates.

most reactions the effect of including substrate concentrations was virtually negligible, as demonstrated by $\rho_{es}$ values far below one (Figure 5C), leading to the conclusion that further regulatory mechanisms must be invoked to explain the observed changes in flux. These mechanisms could include regulation by metabolites other than the substrate or enzyme modifications such as phosphorylation.

## No evidence for growth-limiting biosynthetic pathway expression

Metabolic pathways that synthesize components used directly for biomass, for example, amino acids, nucleotides, and lipids, are a special class of reactions. They are essential under all the conditions in our study and the flux through these pathways is directly proportional to the growth rate, assuming no synthesis/degradation cycles as well as similar biomass composition (Varma and Palsson, 1993). Therefore, if enzyme concentration is limiting for flux through one of these pathways, then it is in turn limiting for growth. We examined the expression of 193 enzymes involved in biosynthesis of amino acids, nucleotides, or cell-wall components as a function of the growth rate. The mRNA signal, which corresponds to the specific mRNA abundance as a fraction of total mRNA, showed good correlation and proportionality with the growth rate for many genes (79 genes with $R^2 > 0.65$ and 21 of these with $\rho_h$ between 0.8 and 1.2), likely due to highly growth rate-dependent $\sigma^A$-based transcriptional regulation (Nicolas *et al*, 2012). However, the protein concentration, calculated as before from RNA concentration and dilution by cell division, did not change proportionally to the growth rate (4 genes with $R > 0.8$ and none with $\rho_h$ between 0.8 and 1.2). This is consistent with the results of a detailed analysis of a constitutively expressed protein in *E. coli* (Klumpp *et al*, 2009). The major reason is that while total mRNA increases with the growth rate, it does not keep up with protein dilution by growth. In sum, we find no evidence that expression of any enzymes involved in biomass component biosynthesis is limiting for growth under unrestricted batch growth conditions in *B. subtilis*. Despite the fact that expression of many of these enzymes is regulated by transcription factors that sense demand for the end product, it appears that the enzymes are present in excess during conditions when they are required.

## On/off reactions are characterized by large transcriptional changes

So far we have examined flux through central metabolic enzymes, which are essential under most growth conditions, and found that enzyme concentration changes were generally much smaller than flux changes. However, for reactions that are completely unnecessary in a subset of conditions, one may expect a clearer correspondence between enzyme concentration and flux. The clearest examples of these are reactions involved in substrate uptake, which are active under only one or two of our conditions. When the expression of proteins involved in uptake systems was examined across conditions, virtually all of them were expressed at much higher levels when the substrate was present compared with other

conditions when they were not required (Figure 6). The magnitude of these changes in expression, which was up to 100-fold in several cases, was far higher than the expression changes seen for most central metabolic enzymes.

Two central metabolic enzymes that did show such large fold changes in expression were the gluconeogenic enzymes GapB and PckA, which are also necessary only under a subset of conditions. These two genes are the only protein-encoding targets of the transcription factor CcpN (Servant *et al*, 2005), which represses their expression during growth on glycolytic substrates. It was previously shown that deletion of *ccpN* leads to a significant growth defect on glucose, due to a futile cycle involving PckA that drains TCA cycle intermediates and a block in upper glycolysis from gluconeogenic GapB activity (Tännler *et al*, 2008b). These effects could be suppressed by deletion of *pckA* or *gapB*, respectively, while constitutive expression of GapB led to blockage in upper glycolysis even in the wild-type background. Those findings, combined with the expression data in this work make it clear that the existence of gluconeogenic flux through GapB is controlled by its transcription, mediated by CcpN. However a more nuanced interpretation, based on the fact that GapB levels did not change significantly between the three gluconeogenic conditions despite significant flux changes, is that once GapB is present at induced levels, the enzyme is available in excess and its precise concentration does not control flux.

Other than GapB and PckA, central metabolic enzymes that were not required in a subset of conditions did not show large expression changes (Figure 6). These included, for instance, PfkA and Fbp, which catalyze the forward and reverse reactions from fructose-6-phosphate to FBP, respectively. Since both enzymes are present at roughly constant levels across all conditions and a futile cycle through these two enzymes would be a major source of ATP dissipation, the activity of these enzymes is likely tightly controlled by other

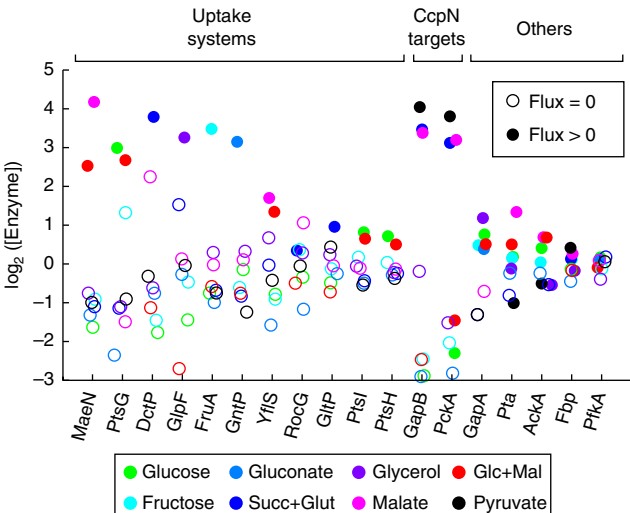

**Figure 6** Enzyme concentrations (inferred from transcripts) for reactions with zero flux under some conditions. All concentrations are relative to the mean across conditions. Solid circles represent conditions with a significant flux ($> 0.1$ mmol gcdw$^{-1}$ h$^{-1}$) while open circles represent conditions with zero flux. For GapA-GapB and Fbp-PfkA, we assumed futile cycle flux to be zero—the actual value cannot be estimated from the available data.

mechanisms. In fact, many metabolites, such as AMP, ATP, PEP, and FBP, have been reported to regulate the flux through this reaction (Fujita and Freese, 1979; Fujita *et al*, 1998).

## Discussion

In this study, we provide evidence that most central metabolic enzymes are available in excess, meaning that changes in central metabolic fluxes are not primarily realized through transcriptional regulation by modification of enzyme concentration. This evidence comes from the fact that fluxes and enzyme concentrations (inferred from transcripts) in central carbon metabolism do not change proportionally across conditions. While previous analysis commonly showed a mismatch between flux and enzyme changes when considering two conditions at once (Daran-Lapujade *et al*, 2007; Schilling *et al*, 2007), by considering as many as eight diverse environmental conditions simultaneously, we are able to assess more generally whether a particular reaction is likely to be limited by enzyme concentration or regulated by other means throughout the diverse environments a microbe may encounter. In all cases, we found that enzyme concentration changed less than flux, that is, we did not find a single reaction where we estimated $\rho_h > 1$. This corresponds to a situation in which enzymes are generally available in excess. The most striking examples of this phenomenon were in the pentose phosphate pathway, where most enzyme concentrations were essentially constant ($<50\%$ variation across conditions) despite 20-fold or more variation in fluxes.

This finding is somewhat surprising in light of the fact that cells clearly face constraints on the amount of protein they can produce (Scott *et al*, 2010), and also in light of analysis showing that protein cost seems to be a major factor even in the choice of the major metabolic pathways available to the cell (Flamholz *et al*, 2013). Nevertheless, several suggestions have been proposed for why cells might choose to keep enzymes available at these overabundant levels (Kochanowski *et al*, 2013a). One appealing motivation is the opportunity to change fluxes quickly, as allosteric regulation or enzyme modifications may act on time scales much faster than gene expression (Xu *et al*, 2012). A second explanation may be the fact that cells constantly have to deal with stochastic fluctuations in gene expression (Raj and van Oudenaarden, 2008). If a key metabolic flux is lowered by a fluctuation in enzyme level, then it is likely to be highly deleterious to the cell, making it more appealing to keep a buffer of enzyme expression. A related point is that it may simply be very difficult to engineer extremely precise regulation of enzyme expression in response to the various conditions that require changes in flux (Price *et al*, 2013), and thus the cell may generally overshoot for expression of key enzymes. Finally, high enzyme concentrations may help the cell control metabolite levels and avoid accumulation of toxic intermediates (Fell and Thomas, 1995; Bar-Even *et al*, 2012).

One pathway in which flux and enzyme concentration changes were approximately proportional across conditions was the TCA cycle. If TCA cycle fluxes are in fact regulated by enzyme levels, then this suggests that the cell tries strongly to minimize excess production of these enzymes, perhaps

because producing them entails high metabolic resource or energy costs. This would essentially mirror the hypothesis that many bacteria choose overflow metabolism at high growth rates in place of the more efficient respiratory metabolism precisely to avoid the high cost of expression of TCA cycle and respiratory chain enzymes (Molenaar *et al*, 2009). Transcriptional control of TCA cycle fluxes would also be consistent with earlier findings that only partitioning of fluxes into the TCA cycle, but not other pathways, is affected by transcription factor deletions in *E. coli* and *S. cerevisiae* (Fendt *et al*, 2010; Haverkorn van Rijsewijk *et al*, 2011).

An alternative interpretation of the observed high $\rho_h$ values in the TCA cycle is that even though enzyme levels may not be limiting, the cell increases them precisely in order to maintain metabolite homeostasis. Such a strategy could be accomplished via feedback from substrate or product levels and would parallel one strategy of engineering increased flux though a pathway without perturbing other parts of metabolism (Kacser and Acerenza, 1993). However, the relatively high changes in metabolite concentrations among TCA cycle intermediates argue somewhat against this interpretation.

Our findings should not be misconstrued as suggesting that transcriptional regulation has no role in regulation of metabolism. In fact, we suggest quite the opposite for reactions that undergo very large fold changes in flux (essentially from zero to non-zero) between our conditions. The most obvious reactions falling into this category are those involved in substrate uptake. Almost all of these genes exhibited large (over 10-fold) changes in inferred protein concentration between conditions where they were required and others where they were not. Similar effects were seen for two enzymes, GapB and PckA, that were required under gluconeogenic but not under glycolytic conditions. However, among the three gluconeogenic conditions, there was virtually no change in enzyme expression despite significant changes in flux, suggesting that when these enzymes are necessary they are present in excess.

With the exception of the two examples discussed above, even large changes in flux through enzymes in central metabolism were not caused by the corresponding changes in enzyme concentration. If central metabolic enzymes are generally in excess as we suggest, then the next most intuitive picture is that flux is regulated far upstream, such as at the level of uptake, and propagates through mass action through the available enzymatic reactions. In this scenario, substrate concentration changes would be the primary drivers of flux changes. Surprisingly, we find that including the effects of substrate concentration changes (based on a simplified kinetic model) is still, in general, insufficient to explain the changes in metabolic fluxes across conditions. The combination of substrate and enzyme changes was able to perfectly explain only a handful of fluxes in central metabolism, most notably the fluxes through the reversible glycolytic enzymes under gluconeogenic conditions, while the remainder of the reactions mostly showed flux changes that were much larger than the combination of enzyme and substrate changes.

This result demonstrates that the central fluxes are also not driven by substrate concentration, but instead must be controlled by another mechanism. The two most likely such mechanisms are protein modifications such as

phosphorylation or acetylation, and allosteric regulation of enzymatic activity by non-substrate metabolites. In fact, a study of the *B. subtilis* phosphoproteome showed that glycolytic enzymes are significantly enriched for *in vivo* phosphorylation (Macek *et al*, 2007). However, the known instances of phosphorylation still only account for a fraction of the reactions we analyzed. Acetylation, which was recently shown to have a major role in regulation of *S. enterica* metabolism (Wang *et al*, 2010), represents another intriguing possibility, particularly since it could be dependent on the concentration of the key central metabolic intermediate acetyl-CoA.

Nevertheless, it is likely that allosteric regulation of enzyme activity by small metabolites is also ubiquitous and important for regulation of fluxes. High-throughput studies aimed at mapping enzyme–metabolite interactions have suggested that only a tiny fraction ($<10\%$) of these interactions have been characterized (Gallego *et al*, 2010; Li *et al*, 2010). It is likely that the true interaction network of enzymes and metabolites is dense and filled with possibly weak interactions that nevertheless exert a degree of control over metabolic fluxes (Rabinowitz *et al*, 2008; Goyal *et al*, 2010). This category would also include product inhibition, which is thought to affect a large number of reactions (Fell and Thomas, 1995).

We have introduced a framework for the analysis of transcriptomic, metabolomic, and fluxomic data sets to deduce the regulation likely to be responsible for the modification of cellular metabolism. Such data sets are becoming more commonly available due to advances in experimental techniques, but methods to integrate different types of data are lacking. By using an extended version of regulatory analysis, we are able to state with high confidence that most reactions in central metabolism are not controlled by transcriptional regulation in response to significant environmental perturbations. Future advances in understanding the regulation of metabolic fluxes are likely to come from more careful examination of the relationship between fluxes and metabolite levels, aided by metabolomics methods that can concurrently quantify large numbers of different compounds (Fuhrer *et al*, 2011; Baran *et al*, 2013), as well as from high-throughput methods to detect post-transcriptional modifications (Bodenmiller *et al*, 2010).

# Materials and methods

## Strains and growth conditions

All experiments were performed with *B. subtilis* BSB168, a prototrophic derivative of *B. subtilis* 168 *trpC2* (Büscher *et al*, 2012). For each growth experiment, frozen glycerol stocks were inoculated into LB medium and after 5 h of growth, diluted into 5 ml of M9 medium with appropriate carbon source. After overnight growth to OD600 between 0.5 and 1.0, the cultures were again diluted into 30 ml of fresh M9 medium in 500 ml non-baffled shake flasks. All cultivation was done at 300 r.p.m. and 37°C.

The M9 minimal medium consisted of the following components (per liter): 8.5 g $Na_2HPO_4 \cdot 2H_2O$, 3 g $KH_2PO_4$, 1 g $NH_4Cl$, 0.5 g NaCl. The following components were sterilized separately and then added (per liter of final medium): 1 ml 0.1 M $CaCl_2$, 1 ml 1 M $MgSO_4$, 1 ml 50 mM $FeCl_3$ and 10 ml trace salts solution. The trace salts solution contained (per liter): 170 mg $ZnCl_2$, 100 mg $MnCl_2 \cdot 4H_2O$, 60.0 mg $CoCl_2 \cdot 6H_2O$, 60.0 mg $Na_2MoO_4 \cdot 2H_2O$ and 43.0 mg $CuCl_2 \cdot 2H_2O$. Filter-sterilized carbon sources were added separately to the medium, pH neutralized with 4 M NaOH where necessary. For $^{13}$C-labeling experiments, the same final concentrations were used, but the carbon source was added directly to the shake flask as a mixture of 20% (w/w) uniformly labeled carbon source ($>98\%$ isotopic purity) and 80% (w/w) naturally labeled carbon source.

## Physiological parameters

Extracellular substrate and byproduct concentrations were measured by HPLC analysis using an Agilent 1100 series HPLC stack (Agilent Technologies, Waldbronn, Germany) in combination with an Aminex HPX-87H polymer column (Bio-Rad, Hercules, CA, USA). Sugars were detected with a refractive index detector and organic acids with an UV/Vis detector. Substrate or product yields were calculated by linear regression of external concentration against biomass, and specific rates were calculated as yield multiplied by the growth rate. At least five time points during the exponential growth phase were used for the regression analysis. Cell growth was monitored photometrically at 600 nm and cell dry weight was inferred from a predetermined conversion factor of 0.48 g cells/OD$_{600}$ (Tännler *et al*, 2008a). All measurement errors for physiological parameters are reported as the standard deviation of 2–3 biological replicates.

## Metabolic flux analysis

Biomass sample processing and GC-MS analysis to determine isotopomer fractions of proteinogenic amino acids was performed as previously described (Zamboni *et al*, 2009). Stoichiometric network models were based on a core model containing the reactions of central carbon metabolism (Oh *et al*, 2007). When unconstrained by labeling information, futile cycle fluxes were set to zero. The growth rate-dependent biomass requirements of *B. subtilis* were previously established (Dauner *et al*, 2001) and added to the network as unidirectional biomass precursor withdrawing reactions. Metabolic fluxes were derived using the whole isotopomer modeling approach (Van Winden *et al*, 2005). The procedure uses the cumomer balances and cumomer to isotopomer mapping matrices (Wiechert *et al*, 1999) to calculate the isotopomer distributions of metabolites in a predefined stoichiometric network model for a given flux set. The flux set that gives the best correspondence between the measured and simulated 13C-label distribution is determined by non-linear optimization and denoted as the optimal flux fit. All calculations were performed in Matlab 7.6.0 (The Mathworks Inc, Natick, MA, USA).

## Transcriptome profiling

Sample collection and RNA purification were performed as previously described (Eymann *et al*, 2002). Three replicates from independent cultures were done for each condition. Synthesis of labeled cDNA, array hybridization, and signal acquisition was carried out by Nimblegen using tiling arrays consisting of 383 149 isothermal probes covering the entire genome of *B. subtilis* (GenBank: AL009126) (Nicolas *et al*, 2012). Signals from the 24 chips were scaled by quantile-quantile normalization (similar results were also obtained without q-q normalization; Supplementary Figure S4). For each condition, one of the three data sets was eliminated based on its Euclidean distance from the other two data sets corresponding to the same condition. The remaining data were used to calculate a mean and standard deviation for each condition.

## Sampling and extracting for metabolite quantification

Two samples for intracellular metabolite quantification were taken within 5 min of each other from the shake flask cultures during exponential growth at an OD$_{600}$ between 0.8 and 1.2. In all, 2 ml of culture was vacuum filtered on a 0.45-μm pore size nitrocellulose filter

(Millipore) and immediately washed with two volumes of fresh M9 medium containing the respective carbon source and adjusted to the pH of the culture at the time of sampling. Sampling was performed in a room kept at 37°C. After washing, the filter was directly transferred for extraction into 4 ml of 60% (v/v) ethanol/water and kept at 78°C for 2 min. The metabolite extract was separated from cell debris and nitrocellulose by centrifugation at $14\,000\,g$ at 4°C for 10 min. The supernatants were dried at 0.12 mbar to complete dryness in a speed vac set-up (Christ, Osterode am Harz, Germany). Dry metabolite extracts were stored at $-80$°C until analysis.

Metabolite concentrations were determined by using an ion-pairing ultrahigh performance liquid chromatography-tandem mass spectrometry method (Büscher *et al*, 2010). Dry metabolite extracts were resuspended in 100 μl, 10 μl of which was injected on a Waters Acquity UPLC with a Waters Acquity T3 end-capped reverse phase column ($150 \times 2.1\,mm \times 1.8\,\mu m$; Waters Corporation, Milford, MA, USA). Metabolites were detected on a tandem mass spectrometer (Thermo TSQ Quantum Triple Quadropole with Electron-Spray Ionization; Thermo Scientific, Waltham, MA, USA).

### Estimation of $\rho_h$ and calculation of *P*-values

To calculate the best linear fit while allowing for outliers, we use the weighted Theil-Sen estimator (Jaeckel, 1972; Birkes and Dodge, 1993). We calculate the slope for each pair of points $\{(x_i, x_j),(y_i, y_j)\}$ and the estimated slope is the weighted median of the pairwise slopes, using weights proportional to the distance $(x_i - x_j)^2$. To estimate $\rho_h$ while obtaining a confidence interval and *P*-value, we obtain a distribution of slopes by performing the fitting multiple times ($N = 2000$), each time sampling a random subset of the data and concurrently perturbing the data by adding Gaussian noise with standard deviation given by the estimated measurement errors. The reported $\rho_h$ is the median of this distribution, and the 95% confidence interval and the *P*-value for $\rho_h > 1$ are obtained directly from the distribution. Additional analysis of $\rho_h$ values under random permutation of conditions is shown in Supplementary Figure S5.

### Supplementary information

### Acknowledgements

We thank members of our laboratories, particularly L Gerosa and M Zampieri, for helpful discussions. Funding for this work was provided by the European Commission projects BaSysBio (LSHG-CT-2006-037469) and BaSynthec (FP7-244093) as well as the German Research Foundation (DFG).

*Author contributions:* VC performed the data analysis and integrated the different data sets, with input from MU and supervision from JS. LLC carried out the microarray experiments. RJK performed the growth, physiology, and $^{13}$C flux analysis. HL performed the metabolomics experiments and quantification. MJ and SA supervised the transcriptomics work and analysis. US, JS and SA conceived the study. US oversaw the research. VC and US wrote the manuscript. All authors have read and approved the manuscript.

### Conflict of interest

The authors declare that they have no conflict of interest.

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
