## [Review Process File · Molecular Systems Biology]

Transcriptional regulation is insufficient to explain substrate-induced flux changes in *Bacillus subtilis*

Victor Chubukov, Markus Uhr, Ludovic Le Chat, Roelco Kleijn, Matthieu Jules, Hannes Link, Stephane Aymerich, Jörg Stelling, Uwe Sauer

Corresponding author: Uwe Sauer, ETH

Review timeline:

Submission date:	15 June 2013
Editorial Decision:	15 July 2013
Revision received:	29 July 2013
Editorial Decision:	03 September 2013
Revision received:	16 October 2013
Accepted:	23 October 2013

Editors: Maria Polychronidou, Thomas Lemberger

Transaction Report:

1st Editorial Decision

15 July 2013

Thank you again for submitting your work to Molecular Systems Biology. We have now heard back from the two referees who agreed to evaluate your manuscript. As you will see from the reports below, the reviewers acknowledge that the study presents interesting findings. However, they raise a series of concerns and make suggestions for modifications, which should be carefully addressed in a revision of the manuscript. Without repeating all the points listed below, most of the reviewers' comments refer to the need to provide additional explanations and clarifications regarding the analysis and interpretation of the results.

On a more editorial level, we would like to kindly ask you to deposit the generated datasets in the appropriate public databases or in the Supplemental Information. (Additional information is available in the "Guide for Authors" section in our website.)

Furthermore, we would like to ask you to include the links and accession numbers in the "Data Availability" section of your manuscript.

REFeree REPORTS

Reviewer #1:

The manuscript aims to quantify the contribution of different governing forces to the regulation of metabolic flux in the central carbon metabolism of *Bacillus subtilis*. The main question addressed is

whether gene expression data have sufficient predictive power to explain the variance in metabolic flux. To address this issue the authors use simultaneous measurements of gene expression data and metabolic flux data taken from 8 cultures of *Bacillus subtilis* grown on various carbon sources. Using a simple mathematical model driven by the rules of enzyme kinetics the authors show that only in a minority of the cases can expression data explain the variance in flux. The authors venture further and establish that metabolite concentrations are also insufficient to explain this variance, concluding that change in flux is in large part due to post translational regulation.

The manuscript is very clearly written and presents an example of good, solid, basic science. Though the result of the paper is negative in nature, the conclusion is sound and significant.

Though I have a few questions and suggestions, I believe that the paper should be published in MSB, in its present form or after minor revision.

Questions to the authors:

1. Have you considered repeating the mathematical analysis without performing quantile normalization? Quantile normalization forces the distribution of expression values across samples to be the same. In my experience, this could prove significant when performing regression analysis, as it distorts the magnitude of the change in expression.
2. Is the raw data used available to the public? Other researchers might wish to repeat the analysis in order to confirm your results.
3. Do the experimental results vary significantly from random? How many reactions are expected to yield high Rho_h values under a random permutation of the expression data?

Reviewer #2:

The experimental data presented in this paper is very interesting, and the overall main conclusion is probably correct. However, this is rather undermined by some loose and superficial reasoning in the examination and interpretation of the results.

The main issue is that hierarchical control analysis is taking a systems-level view of the mechanisms behind changes in metabolism; it is not a method of inferring flux control coefficients, and certainly not 'rate-limitingness'. The use of the rate-limiting term in the manuscript was unnecessary and largely unwarranted, and an immediate improvement would be obtained by its removal.

The simplest argument for this comes from Kacser & Burn's original proof of the flux summation theorem, but is placed more in the context of this manuscript by the Acerenza & Kacser 'Universal method' paper. Essentially, if the amounts of every enzyme in a pathway are increased simultaneously by an amount α , then the flux increases by α and the metabolite concentrations are unchanged. Examination of the flux change and enzyme change at any step would show perfect proportionality, but show nothing about the distribution of control, which in fact has remained unchanged because there is no change in metabolite concentrations (and hence elasticities).

A conclusion about the degree of transcriptional control and the rate-limitingness (or flux control) of an enzyme could only be reached if a single enzyme alone were affected. This is not the case here. In fact, it is very surprising that the authors give no consideration to the fact that some of the transcriptional changes will be highly correlated by virtue of the fact that a number of the enzyme genes occur in operons. This is the case for *TpiA*, *Pgk*, *Pgm* and *Eno*, for example, and indeed the results for these cluster quite closely in Fig. 3. Certainly the last three of these effectively form an enzyme subset, which means that their steady state fluxes are perfectly correlated, and it is obvious that the results for them can contain no information on their relative flux control potential. Looking at it another way, any particular correlation between flux and enzyme transcription could represent an indirect effect of another enzyme that has correlated transcription by mutual operon/regulon membership and/or correlated flux by mutual enzyme subset membership, and these should be explicitly considered.

The other side of this argument, given insufficient weight by the authors, is that control of

metabolism is also about concentrations as well as fluxes, and the more difficult part of this is maintaining metabolite homeostasis. There has long been discussion of this issue, not reflected in this manuscript. Some of it is reviewed in Fell & Thomas (Biochem. J., 311, 35-39 (1995)). A second reason for mentioning this paper is that it gives an example of the need to consider the change in product concentration as well as the substrate concentration in determination whether metabolite concentration changes can account for a change in flux through an enzyme. This issue was glossed over in the manuscript when considering metabolic regulation. It also offers another explanation, not considered by the authors, of why near-equilibrium enzymes without allosteric modifiers might need to be kept at very high levels.

Finally, in the discussion, the authors state that protein modification by phosphorylation could only account for alteration of activity of a small fraction of enzymes. Phosphorylation is not the only possibility; enzyme acetylation may turn out to be significant (Wang et al, Science 327, 1004-7, 2010).

1st Revision - authors' response

29 July 2013

Reviewer #1:

The manuscript aims to quantify the contribution of different governing forces to the regulation of metabolic flux in the central carbon metabolism of Bacillus subtilis. The main question addressed is whether gene expression data have sufficient predictive power to explain the variance in metabolic flux. To address this issue the authors use simultaneous measurements of gene expression data and metabolic flux data taken from 8 cultures of Bacillus subtilis grown on various carbon sources. Using a simple mathematical model driven by the rules of enzyme kinetics the authors show that only in a minority of the cases can expression data explain the variance in flux. The authors venture further and establish that metabolite concentrations are also insufficient to explain this variance, concluding that change in flux is in large part due to post translational regulation.

The manuscript is very clearly written and presents an example of good, solid, basic science. Though the result of the paper is negative in nature, the conclusion is sound and significant.

Though I have a few questions and suggestions, I believe that the paper should be published in MSB, in its present form or after minor revision.

We thank the reviewer for his positive review and kind words.

Questions to the authors:

1. Have you considered repeating the mathematical analysis without performing quantile normalization? Quantile normalization forces the distribution of expression values across samples to be the same. In my experience, this could prove significant when performing regression analysis, as it distorts the magnitude of the change in expression.

This is a good suggestion, and we have repeated the analysis without quantile-quantile normalization. The effects were essentially negligible, as the normalization affected only the extreme tails of the distribution, where few of the metabolic enzymes analyzed here were found. The only enzyme where an effect was seen was PfkA due to a change in the calculated expression in the gluconate condition (this resulted in a large change in ρ_h since only three datapoints were available for this enzyme). The results of the analysis are shown in Figure S4.

2. Is the raw data used available to the public? Other researchers might wish to repeat the analysis in order to confirm your results.

We strongly believe in making data available to the public and we are working with the MSB editors to make sure that all data is easily accessible. The microarray data is already available online, while the flux and metabolite data is provided as supplementary tables.

3. Do the experimental results vary significantly from random? How many reactions are expected to yield high ρ_h values under a random permutation of the expression data?

We have performed this analysis and the results are shown in Figure S5. Even under randomly permuted expression values, many enzymes never achieve a high ρ_h simply due to the fact that enzyme expression changes were of a much lower magnitude than flux changes. However, for those enzymes where enzyme and flux changes had similar orders of magnitude, the true ρ_h was almost always far higher than expected by chance.

Reviewer #2:

The experimental data presented in this paper is very interesting, and the overall main conclusion is probably correct. However, this is rather undermined by some loose and superficial reasoning in the examination and interpretation of the results.

We appreciate that the reviewer finds the work interesting and agrees with our conclusion. We believe the data support the statements we have made, and we have attempted to clarify our reasoning and terminology in this revision.

The main issue is that hierarchical control analysis is taking a systems-level view of the mechanisms behind changes in metabolism; it is not a method of inferring flux control coefficients, and certainly not 'rate-limitingness'. The use of the rate-limiting term in the manuscript was unnecessary and largely unwarranted, and an immediate improvement would be obtained by its removal.

We agree completely that we are taking a systems-level view of the mechanisms behind changes in metabolism; in fact we thank the reviewer for this nice wording, which we have adopted verbatim in the introduction. We have also attempted to make clearer that we are looking at large changes in cellular metabolism, as compared to flux control coefficients, which are typically discussed in the context of infinitesimal changes away from a known steady state. The relation of ρ_h to rate-limitingness occurred only in one instance, and we have modified the text to remove this controversial term.

Along with changes to the wording throughout the text, we have added the following paragraph at the end of the section "Extension of regulatory analysis to quantify contribution of transcriptional changes to flux" (page 5):

"While in the limiting case of small perturbations, ρ_h is equivalent to the flux control coefficient (a measure of the flux change upon a perturbation in a single enzyme level (Small & Kacser, 1993; Sauro *et al*, 1987; Fell, 1998)) it should not be thought of as an estimate of the flux control coefficient in general. Instead, it is a measure of the contribution of transcriptional regulation in realizing the large changes in metabolic operation that take place upon significant environmental perturbations. Another caveat is that even in the case of $\rho_h \approx 1$ mechanisms other than transcriptional regulation may still play a large role in controlling flux, while transcriptional regulation matches flux changes by chance, though this is not generally likely. However, in the case of $\rho_h < 1$, we can definitively conclude that transcriptional control cannot fully explain the data."

The simplest argument for this comes from Kacser & Burn's original proof of the flux summation theorem, but is placed more in the context of this manuscript by the Acerenza & Kacser 'Universal method' paper. Essentially, if the amounts of every enzyme in a pathway are increased simultaneously by an amount alpha, then the flux increases by alpha and the metabolite concentrations are unchanged. Examination of the flux change and enzyme change at any step would show perfect proportionality, but show nothing about the distribution of control, which in fact has remained unchanged because there is no change in metabolite concentrations (and hence elasticities).

We do not dispute at all the conclusions of Acerenza & Kacser. In fact, we would claim that what we are examining is precisely the contrapositive of the flux summation theorem. As Kacser shows, if all enzyme levels increase by alpha, fluxes will increase by alpha and metabolites will stay constant. Therefore, if we observe fluxes increasing by alpha, and enzymes increasing by less than

alpha, relevant metabolites did not stay constant and thus either were responsible for some of the flux changes or other regulation mechanisms such as post-translational modifications must be invoked.

A conclusion about the degree of transcriptional control and the rate-limitingness (or flux control) of an enzyme could only be reached if a single enzyme alone were affected. This is not the case here. In fact, it is very surprising that the authors give no consideration to the fact that some of the transcriptional changes will be highly correlated by virtue of the fact that a number of the enzyme genes occur in operons. This is the case for TpiA, Pgc, Pgm and Eno, for example, and indeed the results for these cluster quite closely in Fig. 3. Certainly the last three of these effectively form an enzyme subset, which means that their steady state fluxes are perfectly correlated, and it is obvious that the results for them can contain no information on their relative flux control potential. Looking at it another way, any particular correlation between flux and enzyme transcription could represent an indirect effect of another enzyme that has correlated transcription by mutual operon/regulon membership and/or correlated flux by mutual enzyme subset membership, and these should be explicitly considered.

As the reviewer correctly points out, the fluxes through enzymes in a linear pathway will be identical, and it's not surprising that the transcript levels are highly correlated given the operon structure. However, in some other cases we do observe differences in expression within an operon (also observed in Güell et.al.

10.1126/science.1176951 and Nicolas et.al. 10.1126/science.1206848), so we see no reason not to analyze each enzyme separately. The key point is that we are not trying to make any statement about the relative flux control coefficients of these enzymes – all we claim is that in comparing large changes in metabolic operation, mechanisms other than increase of enzyme concentration must have been responsible for changes in flux. The lower glycolysis pathway mentioned by the reviewer is a perfect illustration of this – all the enzymes changed by approximately the same fraction between conditions, so regardless of their relative control coefficients, the flux should change by the same fraction if nothing else in the system changed. The fact that flux did not change by the same fraction means other important aspects of the system did change.

Again, we have made changes to our wording throughout the text, making it clear that we looking at the mechanisms behind large changes in metabolic operation.

The other side of this argument, given insufficient weight by the authors, is that control of metabolism is also about concentrations as well as fluxes, and the more difficult part of this is maintaining metabolite homeostasis. There has long been discussion of this issue, not reflected in this manuscript. Some of it is reviewed in Fell & Thomas (Biochem. J., 311, 35-39 (1995)). A second reason for mentioning this paper is that it gives an example of the need to consider the change in product concentration as well as the substrate concentration in determination whether metabolite concentration changes can account for a change in flux through an enzyme. This issue was glossed over in the manuscript when considering metabolic regulation. It also offers another explanation, not considered by the authors, of why near-equilibrium enzymes without allosteric modifiers might need to be kept at very high levels.

We agree completely that product concentrations are very likely to affect fluxes. One of the future directions that similar work could take would be searching for likely metabolite regulators of each flux by considering the relative changes in metabolite and flux concentrations for all metabolites, not just substrates. In that context, it would make sense to focus particularly on reaction products. However, for this manuscript, we wanted to keep the focus on the contribution of transcriptional regulation, and thus only addressed the most intuitive type of metabolic regulation: substrate accumulation. We now explicitly mention product inhibition in the discussion on page 12.

We also agree that maintaining metabolite homeostasis, for instance, to avoid toxic intermediates, could be a crucial goal. This is now mentioned in the discussion on page 11.

Finally, in the discussion, the authors state that protein modification by phosphorylation could only account for alteration of activity of a small fraction of enzymes. Phosphorylation is not the only possibility; enzyme acetylation may turn out to be significant (Wang et al, Science 327, 1004-7, 2010).

We agree completely and thank the reviewer for pointing out this reference to us. We have now mentioned this in the discussion (page 12).

2nd Editorial Decision

03 September 2013

Thank you again for submitting your work to Molecular Systems Biology. We have now heard back from the referee who accepted to evaluate the revised study. As you will see, the reviewer is globally supportive and I am pleased to inform you that we will be able to accept it for publication. This reviewer is nevertheless still raising some detailed points that we would kindly ask you to address as well as possible with suitable amendments to the text.

Please resubmit your revised manuscript online, with a covering letter listing amendments and responses to each point raised by the referees. Please resubmit the paper ****within one month**** and ideally as soon as possible. Once we receive your final revision, it will not take long to make a final decision.

Reviewer #2:

In general the authors have moved in the right direction in making improvements in the manuscript. Unfortunately, they still exhibit some misunderstanding of the meaning of p_h and its relationship to the flux control coefficient. In their response, and the new text they include the statements:

"While in the limiting case of small perturbations, p_h is equivalent to the flux control coefficient (a measure of the flux change upon a perturbation in a single enzyme level (Small & Kacser, 1993; Sauro et al, 1987; Fell, 1998)) it should not be thought of as an estimate of the flux control coefficient in general. Instead, it is a measure of the contribution of transcriptional regulation in realizing the large changes in metabolic operation that take place upon significant environmental perturbations. Another caveat is that even in the case of p_h mechanisms other than transcriptional regulation may still play a large role in controlling flux, while transcriptional regulation matches flux changes by chance, though this is not generally likely."

The first sentence is unequivocally wrong. The flux control coefficient is defined in terms of the flux change brought about by a change in enzyme activity, with all other enzymes held constant, hence when we measure it, we know which enzyme is the cause of the flux change. In experiments such as those under consideration, the flux change that goes in to the numerator of p_h is the overall outcome of a number of changes in external metabolites and some number of enzymes, whilst the enzyme change that goes into the denominator has made some uncertain - and in this context unknowable - contribution to the flux change. Hence even in the case of small perturbations, p_h can never equal the flux control coefficient except in the trivial case of metabolic changes caused by targeted perturbation of the amount of a single enzyme. The converse then follows: p_h of an enzyme contains no information about its intrinsic influence on any flux changes, hence undermining their second sentence. Though the authors claim to have accepted my point and removed the reference to 'rate-limitingness', they persist in promoting the same concept in different words in two places on p. 5 and in the Discussion on p. 10.

Let me try to make this point a different way. What is the benefit to a cell of increasing the expression of an enzyme that has a flux control coefficient of zero on the flux that is being changed (by alterations elsewhere in the system)? It is that the complement of making its p_h close to one is that its p_m can be close to zero, i.e. there is no need for any perturbations in any of the metabolites that affect its activity, thus avoiding potential knock-on effects in other areas of metabolism that need to remain unperturbed. The extreme case would be the regulation analysis of the Kacser & Acerenza Universal Method. Here, the flux to a desired product is increased with the rest of metabolism, fluxes and metabolite levels, unchanged. The flux control coefficients of the enzymes

are ignored, and every enzyme that needs to carry an increased flux has its amount increased in proportion. The regulation analysis would show that every enzyme affected would have a p_h of 1 and a p_m of 0, achieving the goal of the method. Can I be any more clear that there is no connection between the value of p_h for an enzyme and its potential for limiting the flux? (And can I remind the authors that the original Kacser & Burns, 1973, paper showed that there is no necessary connection between the flux control coefficient and the degree of saturation - or 'excess capacity' in their terminology - so they should not push their interpretation in that direction either, as in p. 6, para 3 and p. 10, para 3.)

So what do the results mean? At one extreme, if the cell changes every enzyme in proportion to the required flux change ($p_h = 1$) - the Universal Method strategy - there is maximum cost in change in protein expression. At the other, the maximum economy in change in protein expression would correspond to changes only in enzymes with significant flux control coefficients, forcing p_m to 1 for all the enzymes with unchanged expression, implying that there must be perturbations in metabolite levels. These would likely be of greater magnitude than the flux changes (because concentration control coefficients of an enzyme are generally larger than its flux control coefficient). The outcome of these experiments illustrates how the cell balances the demand of these two constraints.

The authors have responded to my point about the operon and subset structure, but made no change in the manuscript. I don't mind; I've given a hint and they can choose to ignore it, though I think they're looking through the wrong end of the telescope. They might like to think about the implications of the analysis of the thermodynamics of the glycolytic pathway by Flamholtz, A et al (PNAS, 2013) in this context however.

2nd Revision - authors' response

16 October 2013

Reviewer #2 (Remarks to the Author):

In general the authors have moved in the right direction in making improvements in the manuscript. Unfortunately, they still exhibit some misunderstanding of the meaning of p_h and its relationship to the flux control coefficient. In their response, and the new text they include the statements:

"While in the limiting case of small perturbations, p_h is equivalent to the flux control coefficient (a measure of the flux change upon a perturbation in a single enzyme level (Small & Kacser, 1993; Sauro et al, 1987; Fell, 1998)) it should not be thought of as an estimate of the flux control coefficient in general. Instead, it is a measure of the contribution of transcriptional regulation in realizing the large changes in metabolic operation that take place upon significant environmental perturbations. Another caveat is that even in the case of $p_h \approx 1$ mechanisms other than transcriptional regulation may still play a large role in controlling flux, while transcriptional regulation matches flux changes by chance, though this is not generally likely."

The first sentence is unequivocally wrong. The flux control coefficient is defined in terms of the flux change brought about by a change in enzyme activity, with all other enzymes held constant, hence when we measure it, we know which enzyme is the cause of the flux change. In experiments such as those under consideration, the flux change that goes in to the numerator of p_h is the overall outcome of a number of changes in external metabolites and some number of enzymes, whilst the enzyme change that goes into the denominator has made some uncertain - and in this context unknowable - contribution to the flux change. Hence even in the case of small perturbations, p_h can never equal the flux control coefficient except in the trivial case of metabolic changes caused by targeted perturbation of the amount of a single enzyme. The converse then follows: p_h of an enzyme contains no information about its intrinsic influence on any flux changes, hence undermining their second sentence.

Though the authors claim to have accepted my point and removed the reference to 'rate-limitingness', they persist in promoting the same concept in different words in two places on p. 5 and in the Discussion on p. 10.

We did in fact err in our relation of ρ_h to flux control coefficients. and the reviewer is right that small perturbations are not the relevant factor. We have reorganized the text around this paragraph and hope that we make it clear that we make no attempt to estimate flux control coefficients. We now write:

“However, $\rho_h=1$ does not guarantee that perturbations in enzyme level will lead to changes in flux, nor does it guarantee that these enzymes are not present in excess. Flux could still be controlled by unmeasured metabolic changes, while transcriptional changes match flux changes either purely by chance or as a means of keeping metabolite concentrations constant across conditions (Kacser & Acerenza, 1993; Fell & Thomas, 1995).”

Let me try to make this point a different way. What is the benefit to a cell of increasing the expression of an enzyme that has a flux control coefficient of zero on the flux that is being changed (by alterations elsewhere in the system)? It is that the complement of making its p_h close to one is that its p_m can be close to zero, i.e. there is no need for any perturbations in any of the metabolites that affect its activity, thus avoiding potential knock-on effects in other areas of metabolism that need to remain unperturbed. The extreme case would be the regulation analysis of the Kacser & Acerenza Universal Method. Here, the flux to a desired product is increased with the rest of metabolism, fluxes and metabolite levels, unchanged. The flux control coefficients of the enzymes are ignored, and every enzyme that needs to carry an increased flux has its amount increased in proportion. The regulation analysis would show that every enzyme affected would have a p_h of 1 and a p_m of 0, achieving the goal of the method. Can I be any more clear that there is no connection between the value of p_h for an enzyme and its potential for limiting the flux? (And can I remind the authors that the original Kacser & Burns, 1973, paper showed that there is no necessary connection between the flux control coefficient and the degree of saturation - or 'excess capacity' in their terminology - so they should not push their interpretation in that direction either, as in p. 6, para 3 and p. 10, para 3.)

As outlined above, the reviewer makes clear that $\rho_h=1$ does not imply a flux coefficient of 1. We are happy to state explicitly in the manuscript that enzymes with $\rho_h=1$ do not necessarily have high flux control coefficients in any of the conditions nor that they are necessarily at full capacity. We simply claim that full capacity for these enzymes is plausible (and in our view likely, but this is clearly subjective). Our major focus, however, is on the fact that most enzymes exhibit $\rho_h < 1$. For these enzymes, full capacity is impossible, and we would claim that it could be concluded that they have control coefficients below 1 (at steady state in most of the conditions analyzed), though we choose to analyze our results in terms of capacity, not in terms of flux control coefficients. Our reading of Kacser and Burns is that while unsaturated enzymes may or may not have a high flux control coefficient, saturation does drive the flux coefficient towards unity as long as thermodynamic constraints are satisfied.

Moreover we would like to emphasize that it is not always necessary to frame metabolic regulation in terms of flux control coefficients. In this case the reviewer is right that ρ_h does not necessarily make us any wiser about the flux control coefficients – but the argument that $\rho_h < 1$ implies that enzyme capacity was not limiting still holds. The reviewer could claim that this simply says that most enzymes have flux control coefficients below 1 – a somewhat trivial result in MCA. However, we find ρ_h below 1 for large sets of enzymes or whole pathways, meaning that it is not just a trivial consequence of shared control.

The scenario outlined by the reviewer, where flux changes are realized by proportional changes in all the enzymes corresponds reasonably well to many people’s intuitive understanding of metabolism, but is just one of the many ways that a new flux distribution could be realized. Which scenario actually occurs in nature is an important question even though, as the reviewer correctly points out, the answer will not necessarily tell us anything about the flux control coefficients. It will, however, tell us whether transcriptional control is plausible, and combined with e.g. metabolomics, it may tell us which regulatory interactions are crucial.

So what do the results mean? At one extreme, if the cell changes every enzyme in proportion to the required flux change ($p_h = 1$) - the Universal Method strategy - there is maximum cost in change in protein expression. At the other, the maximum economy in change in protein expression would

correspond to changes only in enzymes with significant flux control coefficients, forcing p_m to 1 for all the enzymes with unchanged expression, implying that there must be perturbations in metabolite levels. These would likely be of greater magnitude than the flux changes (because concentration control coefficients of an enzyme are generally larger than its flux control coefficient). The outcome of these experiments illustrates how the cell balances the demand of these two constraints.

The authors have responded to my point about the operon and subset structure, but made no change in the manuscript. I don't mind; I've given a hint and they can choose to ignore it, though I think they're looking through the wrong end of the telescope. They might like to think about the implications of the analysis of the thermodynamics of the glycolytic pathway by Flamholtz, A et al (PNAS, 2013) in this context however.

We readily admit two points: a) thermodynamic constraints dictate the ranges of metabolite concentrations at which metabolism can operate, and enzyme concentrations may need to change in order to maintain these metabolite concentrations and b) a number of very interesting results have come from considering the cost of protein expression as the major optimization problem for the cell, and there is no doubt that it is important to some degree. We allude to both of these in the manuscript. However, we also put forth a number of arguments for why protein economy may not be the major driving force in metabolic regulation.